# Valuing Resilience Benefits of Microgrids for an Interconnected Island Distribution System

**Alexandre B. Nassif [1],\***[iD]**, Sean Ericson [2], Chad Abbey [1], Robert Jeffers [2], Eliza Hotchkiss [2] and Shay Bahramirad [1]**

1   LUMA Energy, San Juan, PR 00907, USA
2   National Renewable Energy Laboratory (NREL), South Table Mountain Campus, 15013 Denver West Parkway, Golden, CO 80401, USA
\*   Correspondence: nassif@ieee.org

**Abstract:** Extreme climate-driven events such as hurricanes, floods, and wildfires are becoming more intense in areas exposed to these threats, requiring approaches to improve the resilience of the electrical infrastructure serving these communities. Long-duration outages caused by such high impact events propagate to economic, health, and social consequences for communities. As essential service providers, electric utilities are mandated to provide safe, economical and reliable electricity to their customers. The public is becoming less tolerant to these more frequent disruptions, especially in view of technological advances that are intended to improve power quality, reliability and resilience. One promising solution is state-of-the-art microgrids and the advanced controls employed therein. This paper presents and demonstrates an approach to technoeconomic analysis that can be used to value the avoided economic consequences of grid resilience investments, as applied to the islands of Vieques and Culebra in Puerto Rico. This valuation methodology can support policies to incorporate resilience value into any investment decision-making process, especially those which serve the public interest.

**Keywords:** distribution system reliability; networked microgrids; outage restoration; resilience; smart grid

## 1. Introduction

Electricity is an essential service, and electrical infrastructure is a critical and essential element supporting a productive society [1]. Unfortunately, the electricity grid in some jurisdictions is susceptible to frequent and extended outages due to its physical vulnerability, increased risk of major weather events, and in some cases both. Major weather events are becoming more frequent due to climate change and associated weather patterns, which have become the single main cause of service interruption in many jurisdictions. While electric utilities traditionally have constructed their systems to operate under normal weather [2,3], the emerging need to account for high impact events and resilience has started to drive electric utility design standards, guidelines and best practices [4].

Electric infrastructure is capital-intensive and upgrading all components, from generation to end-use services, to withstand the most extreme weather disruptions is not cost-effective [5]. Targeted strategic investments are necessary. Alternative solutions that fall under the industry moniker of non-wires alternatives (e.g., distributed energy solutions, energy storage) can, and should, be evaluated against traditional solutions through comprehensive cost–benefit analyses. Among the alternative solutions, the infrastructure, technological advances, and control systems that are the underlying foundation of microgrids have been shown to not only be an option that can provide the necessary resilience a community needs, but in a more economical way than traditional solutions [6].

Currently, most electric utilities do not have distinct predictive evaluation metrics included in cost–benefit analysis processes to estimate the resilience value of potential

technology investments [7]. The goals of reliability and resilience are related, in that they both focus on avoiding outages [8]. However, in the case of resilience, these outages may be so rare that investments to avoid their effects may or may not pay dividends in their lifetimes. Similarly, utilities may not have sufficient data on how their system responds to extreme events because they have seen very few such events in recent history. Finally, a resilience valuation differs from a reliability valuation because the consequences of longer-duration outages can scale nonlinearly with the duration of these outages [9]. Therefore, a probabilistic approach to resilience-focused cost–benefit analysis that balances the likelihood that a disruptive event will occur with the consequences, which reflects the scaling of the consequences of outages over time, is needed. Such an approach could offer confidence and guidance for policymakers and regulators who may need to see estimates of resilience value before allowing utility ratepayers to cover the costs of resilience-enhancing investments.

There is a strong theoretical foundation for quantifying the value of a grid resilience investment, despite challenges that remain to instantiate these methods within utility planning practice. Rickerson et al. overview a host of analytical practices to value the resilience impact of distributed energy resources (DERs), along with an evaluation of the established methods using four criteria [9]. The established methods reviewed are the contingent valuation method used by the Interruption Cost Estimator (ICE) tool, the damage cost method within the Federal Emergency Management Agency's Benefit Cost Analysis Toolkit, and the input-output methodology within tools such as IMPLAN and REMI. The synthesis shows that none of these methods capture the benefits of avoiding especially long-duration outages, and that these approaches do not reflect the full scope of benefits that electric utility regulators may value. Reference [10] discusses the theoretical underpinning of several of these methodologies, while providing research directions that may overcome the gaps identified by Rickerson et al. Recent efforts by [11] as well as [12] have proposed a hybrid approach that combines several of the concepts overviewed by [10], and offers a clear path forward to calibrating and validating these models. Case studies that directly compare these various approaches and describe the findings using language and concepts that are readily accessible to electric utility regulators were not found in the literature.

This research work presents a real case study of two islands within a multi-island power system operated by a utility that serves about 1.5 million metered premises, providing electricity to nearly 3.2 M residents. A valuation is presented of the economic impact of avoiding both day-to-day outages (e.g., reliability events) and more extreme hurricanes (e.g., resilience events) on the islands' residents. This valuation serves to establish a resilience investment business case for guiding infrastructure investment to modernize the electric grid. A microgrid sizing exercise is presented, consisting of aggregations of DERs leading up to a nested architecture to leverage renewable generation. Renewables can be integrated not only through utility-scale, utility-owned resources, but also through virtual power plants through renewable procurement processes that are currently open for bids. Improving the economic resilience of island communities by investing in the electric grid is one way to approach the power system and its design, where power system infrastructure, along with control and communication architectures, are designed concurrently to integrate DERs and microgrids and ensure the delivery of power to critical loads, even under extreme events [13]. The proposed concept and resilience valuation approach represents a unique opportunity to develop a system according to these tenets.

The paper is organized into five sections. This introduction has introduced the need for methodologies for the quantification of the resiliency value of grid investments such as microgrids and the application of such methods to practical problems. Section 2 presents the proposed valuation methodology for resiliency-enhancing grid investments. Section 3 describes the Vieques Culebra case study and the microgrid DER sizing methodology and costs. Section 4 presents the conceptual design for the microgrid, considering the microgrid

would be composed of utility-owned and customer owned assets. The authors present conclusions in Section 5.

## 2. Valuation Approach

This section presents an approach to valuing the resilience benefits of microgrids, or other resilience-enhancing investments. The approach is based on a foundation of probabilistic risk analysis [14] and economic theory described by [10], and therefore attempts to address a range of different outage events and their probabilities. The approach consists of three steps: (1) selecting event scenarios to determine the frequency of occurrence for these scenarios, (2) determining outage durations for each scenario, and (3) determining the outage costs given outage durations. Three main outage event categories are established: daily reliability events, major hurricane events, and exceptional hurricane events. For each event category, a baseline outage frequency and duration are established, which represents the "do-nothing" investment alternative. Four alternative approaches to valuing outage costs are presented, which differ based on the complexity and the realism with which they address the temporal dynamics of these costs.

Additional approaches were considered but not utilized for this case study. Methods to value the social benefits of a resilience investment such as the social burden analysis proposed by Jeffers et al. are not directly comparable to the economic methods outlined herein because they attempt to measure human well-being losses that are additive and orthogonal to monetary losses, and were therefore not utilized [6]. Application of these social valuation methods in combination with the economic valuation methods may overcome some of the challenges outlined by [9], since they reflect the additional values of electric utility regulators. Additionally, attribute-based measurement of resilience benefits used within a multi-criteria decision analysis framework as described by [14] was considered for this analysis but was not applied because these attribute-based methods measure the conditions that improve resilience, but do not directly measure the benefits of that improved resilience, as well described by [15].

### 2.1. Frequency of Outage Events

Reliability events are characterized based on historical outage frequency and duration during day-to-day operations. The years in which major or exceptional events occur are not used to establish the baseline for reliability events, because they greatly alter the outage data (e.g., Hurricanes Irma and Maria left the islands without power for 11 months, skewing daily operation data). Table 1 summarizes the outage frequency and total cumulative duration across different outage periods, from 5 min up to 24 h, using nine years of outage data. This work uses reliability metrics SAIDI and SAIFI (system average interruption duration index and system average interruption frequency index).

**Table 1.** Outage Frequency and Cumulative Duration.

| Outage Period | Total Outages | Total Outage Minutes (1000 s) | Average Customer Outages | Average Customer Outage Minutes |
|---|---|---|---|---|
| 5 min–1 h | 13,802 | 414 | 4.63 | 139 |
| 1–2 h | 6938 | 624 | 2.30 | 207 |
| 2–4 h | 5848 | 1019 | 1.82 | 314 |
| 4–8 h | 4793 | 1840 | 1.47 | 560 |
| 8–12 h | 3067 | 1790 | 0.85 | 496 |
| 12–16 h | 236 | 193 | 0.05 | 40 |
| 16–24 h | 444 | 514 | 0.12 | 140 |
| **Total** | **35,129** | **6394** | **11.24 (SAIFI)** | **1895 (SAIDI)** |

The major and exceptional hurricane event frequencies are established based on historic records of hurricanes and the outages they caused for the islands. However, because outage data are not as well established for historic hurricanes compared to reliability events, it is assumed that hurricanes coming within 50 nautical miles of the islands have the potential to cause an outage. Table 2 summarizes the number of hurricanes at various strengths within 50 nautical miles (nm) of the islands in the past 170 years [16].

**Table 2.** Hurricane Count at Various Strengths.

| Maximum Strength within 50 nm | Number of Occurrences over 170-Year Interval | Annual Frequency of Occurrence | Recurrence Interval (Years) |
|---|---|---|---|
| Tropical Depression/Storm | 27 | 0.159 | 6 |
| Hurricane-I | 9 | 0.053 | 19 |
| Hurricane-II | 5 | 0.029 | 34 |
| Hurricane-III | 9 | 0.053 | 19 |
| Hurricane-IV | 3 | 0.018 | 57 |
| Hurricane-V | 3 | 0.018 | 57 |

The following assumptions are made to estimate outage frequencies based on hurricane and strength frequencies:

- Major hurricane outage events and extreme hurricane outage events are caused by category III or higher hurricanes.
- Every hurricane category III or higher within 50 nm of the islands causes an outage.
- One in three category IV or V hurricanes that come within 50 nm of the islands cause extreme disruptions.

These assumptions result in the following frequencies of hurricane events:

- Major hurricane outage events occur once every 13 years (annual probability = 7.65%).
- Extreme hurricane outage events occur once every 85 years (annual probability = 1.18%).

### 2.2. Duration of Outage Events

For reliability events, outage durations are established based on historic data as indicated in Table 2. Figure 1 shows that reliability events commonly cause outages lasting less than four hours.

For major and exceptional hurricane outage events, outage durations must be assumed using inference from the available historic data. First, the most recent exceptional hurricane outage event is examined. A resilience curve is established that quantifies the baseline performance in the two smaller islands to past hurricane outage events. The rate of recovery to a loss suffered by the system during *black sky days* (i.e., following a major outage) is a precursor metric to evaluate historic performance. This can be obtained from the difference between the outage rate $O(t)$ and the recovery rate $R(t)$ at time $t$. These instantaneous quantities can be defined as

$$O(t) = OOS_{component}(t), \tag{1}$$

$$R(t) = R_{component}(t), \tag{2}$$

where OOS stands for a component that is Out Of Service. These can be assigned to the number of outages, the number of unserved customers, or another pertinent parameter. As an example, cumulatively, these can be defined as:

$$O(t_i) = \sum_{ti=t0}^{t1} OOS_{customers}(t_i), \tag{3}$$

$$R(t_i) = \sum_{ti=t0}^{t1} R_{customers}(t_i), \tag{4}$$

where these equations represent the customer outages following an event and those recovered, both at time $t_i$. The difference of these two equations is the remaining customers not served.

The metric used to quantify the resilience of the system is the resilience curve, defined as $C(t_i)$, which can be calculated as the difference between customer outages and those recovered: [17]

$$C(t_i) = R(t_i) - O(t_i), \tag{5}$$

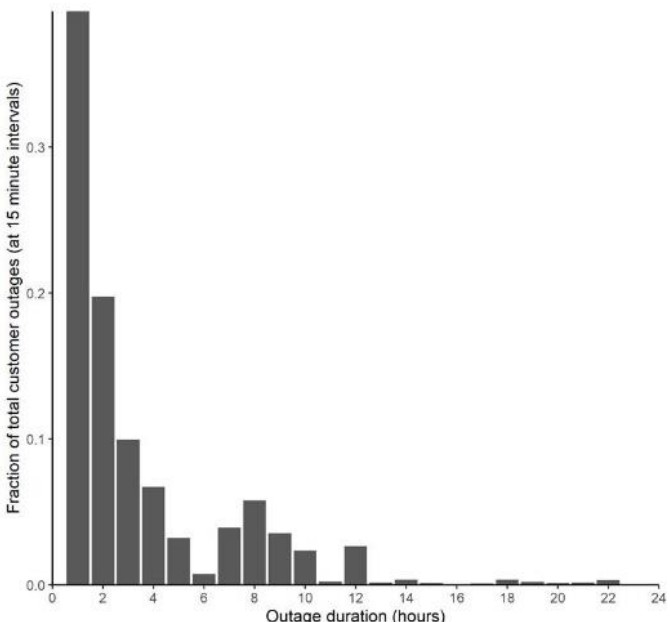

**Figure 1.** Customer outage and duration distribution probability function.

The outage and restoration rates, as well the resilience curves were populated for Vieques and Culebra. The calculation was applied to the summer of 2017 when the system suffered during two category 5 hurricanes. This is the only recent representation of an exceptional hurricane outage event in electric utility data records. The first hurricane (Hurricane Irma) grazed the northeast corner of Puerto Rico, but its wind activity had a significant impact on the smaller islands of Vieques and Culebra. The second hurricane (Hurricane Maria), on the other hand, was devastating, as its path was directly through the center of the main island and over the two small islands, entering the southeast corner and leaving through the northwest. Both of the smaller islands have emergency backup diesel plants to serve the community following these events. In Vieques, the plant was inoperable, and the U.S. Federal Emergency Management Agency (FEMA) hauled in mobile diesel units, as reflected in the outage duration shown in Figure 2, which depicts the outage, restoration, and resilience curves for the timeframe of the two hurricanes and for the two islands combined.

While not all hurricanes will cause similar durations of outages (e.g., the 2017 event resulted in outages of up to 11 months for many residents of Puerto Rico, although electricity was restored to Vieques and Culebra after about 2.5 months), this event provides insight on what could be an upper bound for what might be expected if another category IV or V hurricane damages the islands. Therefore, with the historical data presented, engineers responsible for improving resilience on these islands recommended that the average outage duration for major events be estimated at 1 week on both islands, while the outage duration for exceptional events is estimated at 2 months on both islands. Damages from extreme hurricane events can cause future grid instability. Customer outage minutes were 300% higher the year after Hurricanes Irma and Maria impacted the islands. Based on this,

reliability outages are assumed to increase by 300% the year after an exceptional hurricane event. This assumption is applied in the valuation methodology.

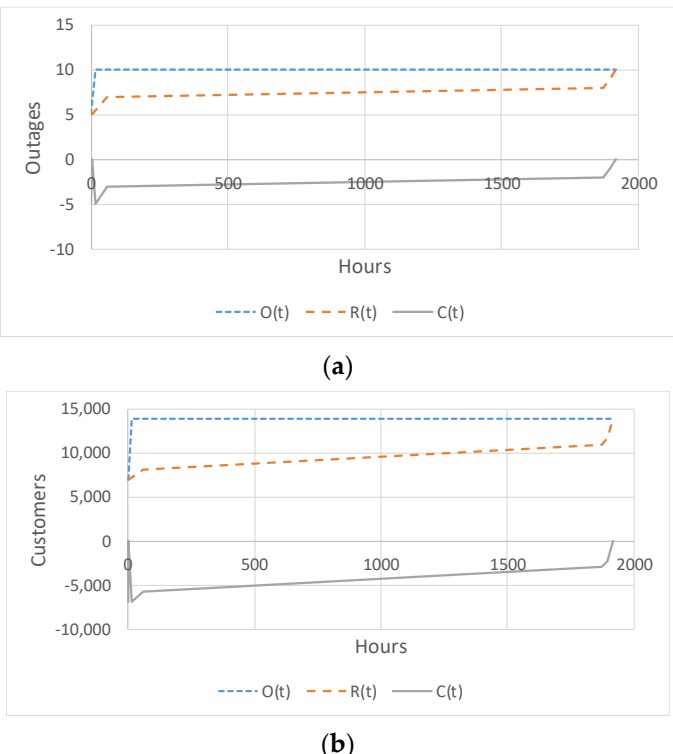

(**a**)

(**b**)

**Figure 2.** Component outage, restoration, and resilience curves for the hurricanes, (**a**) outage count and (**b**) unserved customers.

### 2.3. Baseline Economic Consequence of Outage Events

To apply the valuation methodology to the historic events, a few assumptions were made. The annual value of a resilience upgrade is equal to the expected annual avoided outage costs. This is equal to the sum of costs by outage scenario multiplied by the annual frequency of each scenario. The cost of a given outage scenario is determined by the number and type of customers impacted, as well as the outage duration. Table 3 displays the number of customers by island. The average load across Vieques and Culebra is 5.45 MW.

**Table 3.** Customer Affected in Vieques and Culebra.

| | Number of Customers | | | |
|---|---|---|---|---|
| **Location** | **Residential** | **Commercial** | **Industrial** | **Total** |
| Vieques | 4378 | 465 | 1 | 4863 |
| Culebra | 1121 | 306 | 1 | 1449 |
| Total | 5499 | 771 | 2 | 6312 |

The financial impact on a customer goes well beyond the cost of electricity that was not billed to that customer. In fact, there are different approaches to quantify the value of lost load (VOLL). This section introduces each valuation method's philosophy and results from performing a baseline economic loss calculation across the outage event categories.

The four main economic loss evaluation methods are:

1. A FEMA-assigned cost of USD 174 per person (not customer) per day [18].
2. A jurisdictional survey to obtain a blended cost per lost MWh assigns a cost of USD 31,895 per MWh, as a blended class rate for the islands [19].

3.　Using the Interruption Cost Estimate (ICE) Calculator [20] to estimate a cost of USD 40,430 per MWh, as a blended class rate for the islands.

4.　Using a simple time-varying customer damage function, as described below, to estimate outage costs in each scenario.

Costs from power outages may vary nonlinearly with outage durations. Factors such as food spoilage or limited fuel for backup generators can result in marginal outage costs increasing with outage duration. Meanwhile, mitigation steps can reduce the marginal cost of outages as outage duration increases. Such nonlinearities imply that it may be inappropriate to use a single per MWh value to outage cost. Instead, the per MWh cost of an outage may vary by outage duration.

To incorporate nonlinearities in shorter duration outages, the ICE Calculator was used to estimate outage costs across outage durations from 1 to 16 h, and then multiplied by the frequency of outages of each duration to estimate an average cost of USD 48,940 per MWh, which is 20% higher than the ICE Calculator when average outage durations are used.

The marginal costs (e.g., the additional outage cost per hour after already experiencing an outage of x hours) of long duration outages may be less than, or more than, the marginal costs of shorter duration outages. The results are presented for a range of potential costs for long duration outages. The low-end case sets the average per MWh cost of long duration outages to half of the per MWh cost of short duration outages (USD 24,470/MWh), the mid case sets the two equal (USD 48,940), and the high-end case sets the per MWh cost of long duration outages to double the per MWh cost of short duration outages (USD 97,880).

These numbers result in the costs presented in Table 4, which displays estimates for the annual expected outage costs by event type. These results illustrate the local economic impact of both shorter duration grid outages and the potentially significant cost of low probability but high impact hurricane events.

**Table 4.** Customer Affected in Vieques and Culebra.

| Resilience Valuation Method | Reliability Costs | Major Hurricane Costs | Extreme Hurricane Costs | Total Annual Costs |
|---|---|---|---|---|
| FEMA | USD 46,000 | USD 590,000 | USD 1,284,000 | USD 1,919,000 |
| Surveys | USD 174,000 | USD 2,242,000 | USD 4,884,000 | USD 7,300,000 |
| ICE | USD 220,000 | USD 2,838,000 | USD 6,182,000 | USD 9,240,000 |
| Nonlinear Outage Costs | USD 267,000 | L: USD 1,720,000<br>M: USD 3,440,000<br>H: USD 6,881,000 | L: USD 3,974,000<br>M: USD 7,494,000<br>H: USD 14,535,000 | L: USD 5,961,000<br>M: USD 11,201,000<br>H: USD 21,683,000 |

*2.4. Expected Resilience Value of Avoiding Outage Events*

Table 4 displays the estimated expected annual costs due to power outages, which approximate the maximum potential annual resilience benefits for the microgrid system. Depending on the reliability of the microgrid system, the resilience benefits may only be equal to a fraction of the annual outage costs. Annual resilience values can be converted into present values given information on the relevant discount rates and project lifetime. The resilience value of the microgrid is also only one part of the potential value provided. A full valuation should also include costs and benefits from changes in capital expenditures, fuel costs, and operation and maintenance expenses.

Two assumptions are made to simplify the assessment of resilience benefits for the microgrid:

- The additional cost to operate backup generators to restore power after hurricane events are small in comparison to the outage costs during these events and are therefore omitted.
- The microgrid provides power during all events to calculate the maximum reliability and resilience benefits it could provide. Thus, no outages ever occur with the hardened microgrid.

With these two assumptions the annual resilience benefits are equal to the total annual costs in Table 4.

## 3. Microgrid Case Study

### 3.1. Description of the Islands of Puerto Rico

Within the multi-island case study, the main island of Puerto Rico, where the vast majority of the 2.5 million residents are located, has a 230 kV backbone ring and a meshed 115 kV networked transmission system. The grid system consists of a highly dispersed subtransmission system operating at 38 kV. To note, most industrial and large commercial customers are supplied at the subtransmission level, whereas most commercial and all residential customers are supplied at the distribution level. The utility operates ~1400 distribution feeders, currently supplied at four different voltage levels: 13.2 kV, 8.32 kV, 7.2 kV, and 4.16 kV, which are about 25%, 15%, 3%, and 55% of the feeders, respectively. The two small islands, Vieques and Culebra, which are the subject of this paper, house about 10,000 residents combined. A submarine cable connects one of the 38 kV lines from the main island to one of the two smaller islands, and, in tandem, the second smaller island. These small islands each have a 38/4.16 kV distribution substation, and have three and two distribution feeders, respectively, as illustrated in Figure 3.

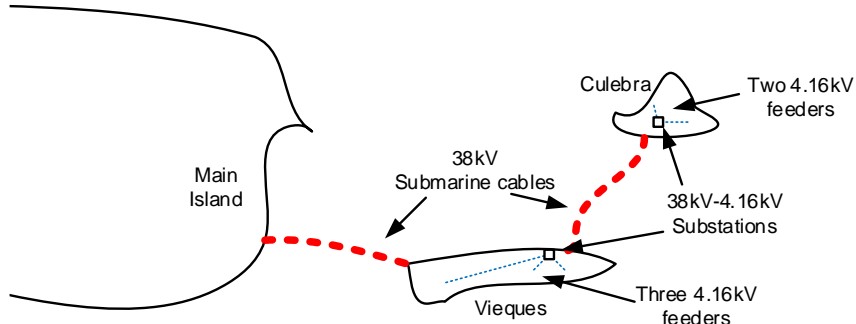

**Figure 3.** A simplified representation of the island system (not to scale).

### 3.2. Sizing and Sensitivity Analysis

Using the valuation analysis, a microgrid was analyzed to determine the different sensitivities. Different objectives were considered when deciding on the sizes of the solar photovoltaic (PV) generation and energy storage to be installed in both small islands. The microgrid optimization tool HOMER was used to size the DERs.

The analysis considered two objectives:

O1:  Reduction in existing diesel consumption when either Vieques or Culebra is separated from the main island grid.

O2:  Reduction in excess energy when Vieques and/or Culebra are separated from the main island grid.

These objectives are considered under four scenarios:

S1:  Vieques is isolated.

S2:  Culebra is isolated.

S3:  Vieques and Culebra are separated from the main island but interconnected to one another.

S4:  Blue sky operation (i.e., both islands are connected to one another and to the main island grid).

To reach a recommendation, a comprehensive search of various microgrid scenarios was conducted. The loading for both islands for one year, using a one-hour resolution, was used. The sizing results are shown in Table 5. These results demonstrate that renewable energy curtailment is unavoidable under practical DER sizes, but scenarios can be optimized to reduce both objectives.

The research conducted to obtain these results also serves as a sensitivity exercise, as it displays the impact of changing the configuration on the diesel consumption, shown in Figure 4. This exercise considers any excess energy that may result from each case. It suggests that both battery energy storage systems (BESS) and PV systems must be increased in size simultaneously to provide significant benefit; otherwise, we observe diminishing returns. This analysis reveals that for the allotted project budget (not discussed in detail in this paper for confidentiality), the proposed size meets the project requirements.

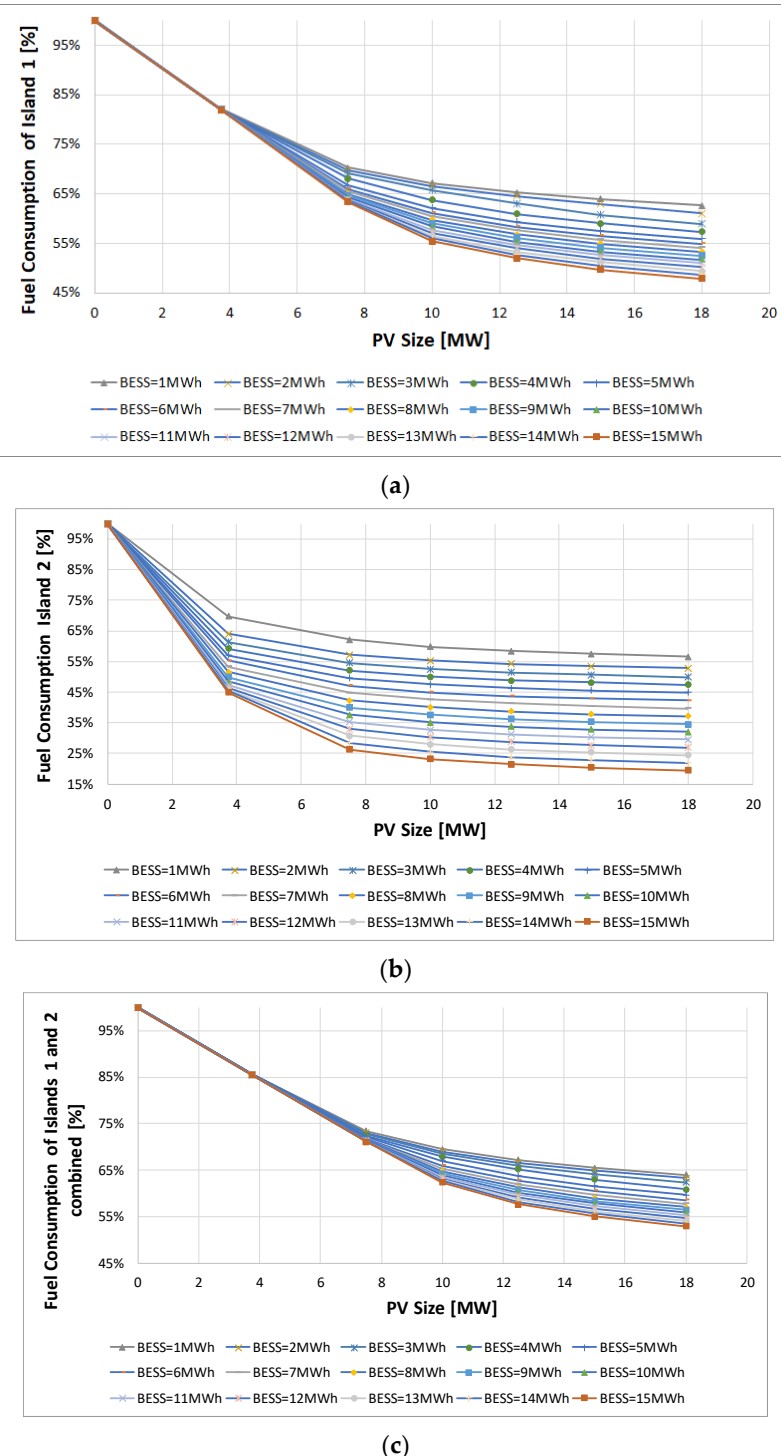

**Figure 4.** Sensitivity analysis results for diesel consumption reduction of different scenarios, (**a**) Vieques microgrid, (**b**) Culebra microgrid, and (**c**) combined Vieques + Culebra integrated microgrids.

The analyses were conducted assuming a total loss of the subtransmission 38 kV supply (black sky event), which results in major PV generation curtailment, as suggested by Figure 5. However, the curtailed PV energy could be used under blue sky operation to further offset the total energy consumption of both islands when the subtransmission supply is available. For blue sky days, this microgrid will result in a substantial positive flow of renewable energy into the main island grid system.

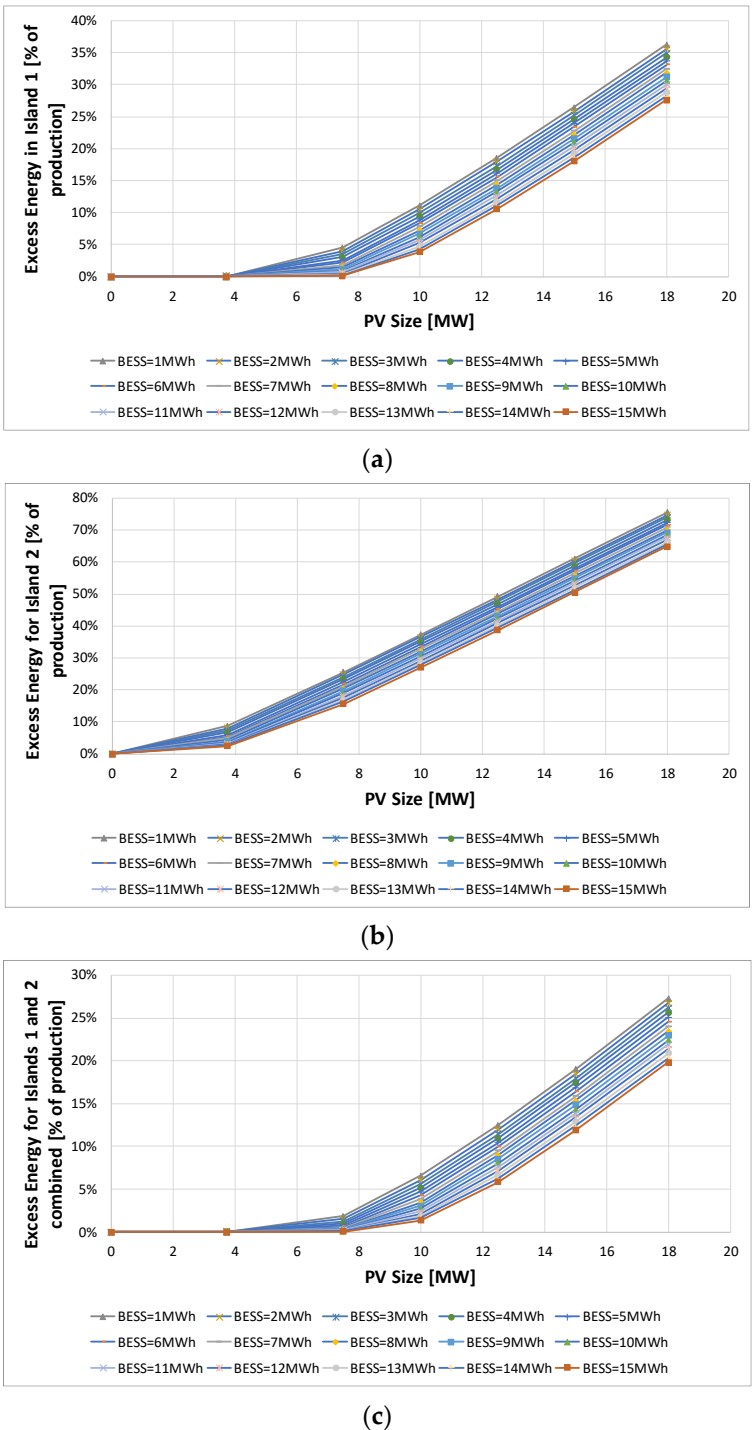

**Figure 5.** Initial feasibility analysis results showing excess energy of different islanded scenarios, (**a**) Vieques microgrid, (**b**) Culebra microgrid, and (**c**) combined Vieques and Culebra integrated microgrids.

**Table 5.** Suggested sizes and their impact on diesel consumption and energy production.

| Scenario | BESS | PV | O1 | O2 | Energy Offset |
|----------|------|------|-----|-----|---------------|
| S1 | 7 MWh | 12 MW | 41% | 13% | 41% |
| S2 | 3 MWh | 3 MW | 35% | 5% | 35% |
| S3 | 10 MWh | 15 MW | 42% | 14% | 42% |
| S4 | 10 MWh | 15 MW | N/A | 0% | 56% |

## 4. Proposed Nested Microgrid Concept

The preliminary microgrid studies suggest a major leap in technology adoption and modernization. The designs are executed so that the resilience valuation presented herein will be realized at least cost to the utility ratepayer. The valuation metrics guide the design, in that they require a highly robust and resilient design to hurricanes to achieve the calculated benefits, and the stated annual expected avoided loss can be a baseline for the cost–benefit comparisons for the design. When planning the execution of this project, it is understood by the authors that it may need to be staged in three phases.

The first phase will focus on delivering resilience benefits quickly by upgrading existing assets and their controls and protections, along with the integration of a microgrid controller to enable island-wide microgrids to provide autonomy to each of the two islands in the event of a power outage. The system will be planned and studied with Phases 2 and 3 in mind to consider how the microgrid controller, protection scheme, communication requirements, and distributed controls will evolve under each of the phases. This staged approach includes energy efficiency measures, the placement of automated switches and sensors, and the development of an evolving operations concept.

Phase 2 will integrate new renewable technologies into the control architecture, developing resilient layers within each microgrid. Finally, Phase 3 will endeavor to operate the two island microgrids in parallel, maximizing benefits under both blue sky and black sky days.

Phases 2 and 3 will enlarge the scope of the microgrids by incorporating a layered community microgrid concept, that considers the behind-the-meter (BTM) microgrid—residential or commercial/facility level—as the fundamental building block of a resilient grid architecture, shown conceptually in Figure 6. Each BTM microgrid is then added to form community microgrids at the level of the service transformer and extended further to the feeder and substation level as part of the substation microgrid commissioned in Phase 1. This system of systems architecture breaks down the complexity of the problem and can be dynamically managed to scale to hundreds of DER assets, including BTM and front-of-the-meter (FTM), utility-owned, or third party owned assets. These DERs can even include mobile assets, such as existing mobile diesel generators or mobile energy storage systems (MBESS) to quickly optimize the system following a major event. All systems are integrated to microgrid controllers by implementing monitoring and controls over a field message bus, with automated discovery and standard data models to favor interoperability and rapid reconfiguration of the system, as illustrated in Figure 7. The island microgrid controller will also provide integration to the SCADA system and can coordinate with back-office applications, such as energy management system (EMS).

Resilient operation of the microgrid following major events will also depend on the availability of both DER resources and communications for data exchange between different microgrid components. The BESS will provide the appropriate energy during and immediately following the event together with the diesel generators, when PV generation may be non-existent or minimal. A couple of hours following the event, available PV generation will then replace the diesel generation and can provide significant diesel offset should outage of the subtransmission connection persist for multiple days or weeks. With regard to communication network resilience, cellular, meshed network and even low-earth orbit (LEO) satellite communication options are being considered for redundant

communication paths. These technical details will need to be fully clarified as part of the design, build and test phases of the project.

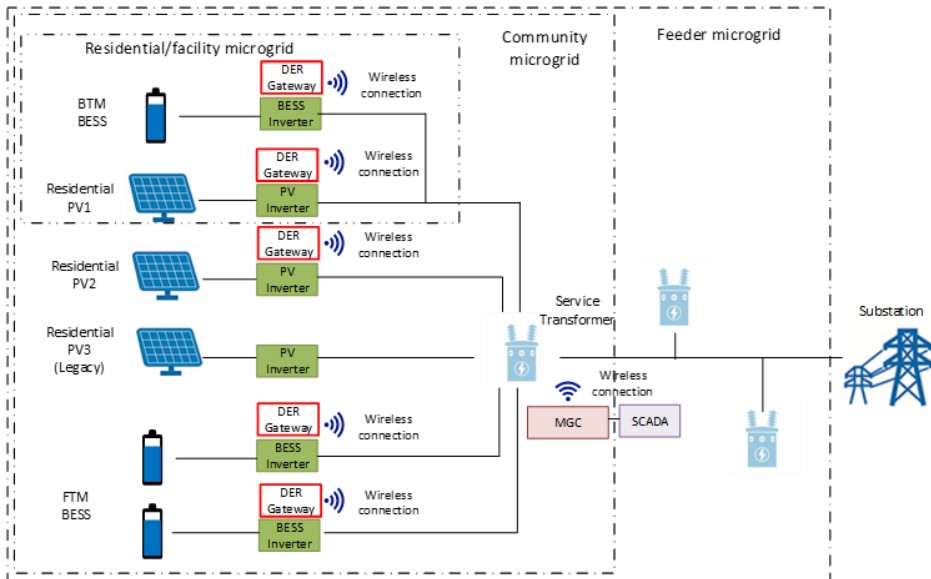

**Figure 6.** Resilient community microgrid layered approach using BTM and FTM DER assets as sublayer of diesel microgrid.

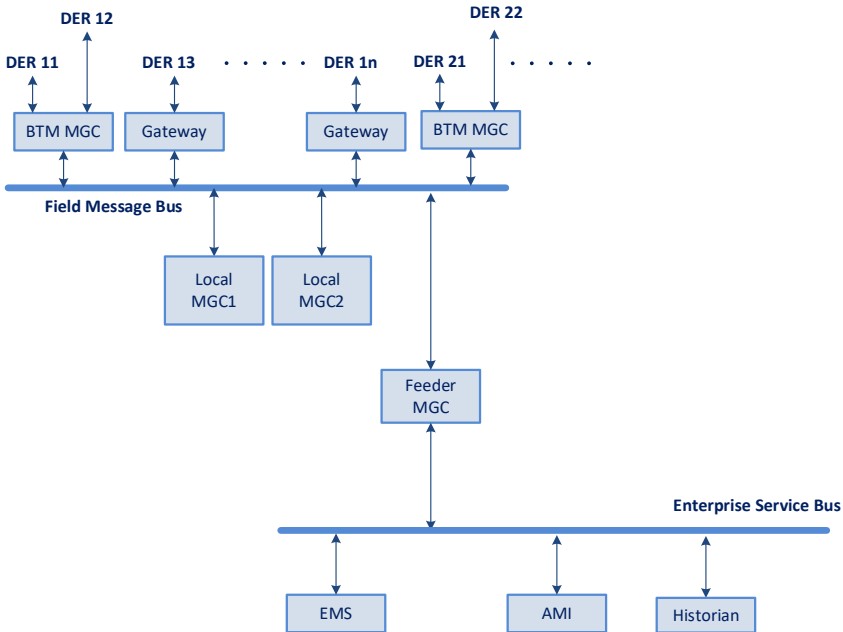

**Figure 7.** Resilient architecture for monitoring and control for the microgrids on both islands.

Based on similar architectures published in [21,22], this project is expected to:

- Improve the resilience of the two islands against extended outages during extreme events.
- Eventually reduce the overall diesel consumption, should the islands experience another extended outage, by more than 45%.
- Reliably operate each of the community microgrid layers (BTM, service transformer, and feeder) in islanded operation, respecting industry standards [23–26].

- Connect and integrate a mobile DER or new DER to the community microgrid within two hours.
- Demonstrate, through the nesting of community microgrids, that the technology can achieve the scalability of more than 500 DER assets, through real-time simulation testing and field validation at the demonstration project.

Finally, as with all critical infrastructure, the microgrid and participating DER will need to be appropriately secured to ensure resiliency to physical or cyberattacks during both normal operation and black sky days. These measures may imply additional costs to the project and will be considered as part of the final design.

## 5. Conclusions

This research work presented the economic resilience performance of a multi-island electric utility during Hurricanes Irma and Maria, calculated for the islands of Vieques and Culebra, through resilience curves and a probabilistic risk-based resilience valuation methodology. The resilience curves suggest it took more than three months to completely restore Vieques. When using IEEE suggested resilience metrics [4], this would result in a storm event X near 100%, which is a total failure. Given how devastating Hurricane Maria was, it is difficult to assert the fragility of the system pre-disaster.

To enhance resilience, a microgrid solution is proposed, the cost of which was measured against the societal impact of the two hurricanes. The proposed microgrids will rely on 10 MWh BESS and 15 MW PV generation and will provide up to a 42% reduction in diesel consumption during loss of the subtransmission supply and up to a 56% reduction in the annual energy consumption of the minigrid of Vieques and Culebra from the main island during blue sky operation.

The economic analysis suggests the the proposed microgrid projects can add between $1.9 and $20 million of expected annual avoided loss by improving the reliability and resilience of the discussed locations, and by supporting the community in future major weather events. The scalable layered architecture under consideration allows scaling to larger systems, or as in the case of the proposed approach, build out resilient sublayers of the diesel microgrid as renewables are integrated through VPP or other energy procurement exercises. As the team plans implementation of the proposed microgrid, it is expected the proposed innovations will benefit underrepresented and vulnerable communities and the industry at large. Furthermore, as the parameters of the resilience valuation methodology are validated, it can serve as a basis for future cost–benefit analysis requirements from policies that encourage electrical system resilience investment.

As shown by the direct comparison within a single case study of four different economic resilience valuation methods, the consideration of non-linear effects due to the compounding economic conditions over the dynamic evolution of especially long-duration power outages leads to a potentially different valuation of resilience benefits for a resilience investment that reduces outage risk. However, this case study did not validate the parameters for the time-varying customer damage function approach, and instead presents three illustrative parameterizations. While theoretically the time-varying customer damage function method is more precise than the three other methods presented, this study shows that obtaining valid parameters setting the shape of these damage functions is critical, since different parameterizations lead to valuations of resilience investments that span a wide range. Future work is necessary to achieve this validation, as outlined by Baik et al. [11]. Furthermore, while a deterministic approach was illustrated to determining the probability of occurrence at different outage durations, in reality these values are highly uncertain. Future work should perform more rigorous uncertainty quantification and sensitivity analysis to describe the range of likely valuations for grid resilience investments.

**Author Contributions:** Conceptualization, A.B.N.; methodology, A.B.N., S.E. and C.A.; software, A.B.N. and S.E.; validation, R.J., E.H. and S.B.; formal analysis, A.B.N. and S.E.; investigation, A.B.N. and S.E.; resources, R.J. and S.B.; data curation, C.A. and E.H.; writing—original draft

preparation, A.B.N. and S.E.; writing—review and editing, A.B.N., S.E., C.A. and R.J.; visualization, E.H.; supervision, S.B.; funding acquisition, R.J. All authors have read and agreed to the published version of the manuscript.

**Funding:** This work was authored (in part) by the National Renewable Energy Laboratory (NREL), operated by Alliance for Sustainable Energy, LLC, for the U.S. Department of Energy (DOE) under Contract No. DE-AC36-08GO28308. This work was supported by the DOE Grid Deployment Office and the Federal Emergency Management Agency. The views expressed in the article do not necessarily represent the views of the DOE or the U.S. Government. The U.S. Government retains and the publisher, by accepting the article for publication, acknowledges that the U.S. Government retains a nonexclusive, paid-up, irrevocable, worldwide license to publish or reproduce the published form of this work, or allow others to do so, for U.S. Government purposes.

**Conflicts of Interest:** The authors declare no conflict of interest.

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
