# Peer review of "Valuing Resilience Benefits of Microgrids for an Interconnected Island Distribution System"

_electronics, doi:10.3390/electronics11244206_

Round 1
Reviewer 1 Report
Table 4 is unfortunately split between two pages. For equation (5), it is not clearly defined what C(ti) is.
Author Response
The authors would like to thank the reviewer for their time reviewing our manuscript. Below is a response to the point(s) raised:
Table 4 is unfortunately split between two pages. For equation (5), it is not clearly defined what C(ti) is.
The authors appreciate the suggestion for improvement and have added text to explain what the resilience curve C(ti) means. The authors also ensured no tables are broken over more than 1 page.
Reviewer 2 Report
In the reviewer’s opinion, the paper could have been more interesting and better organised. In general, the overall contribution remains scientifically poor and technically questionable. In more detail, the paper’s title is quite clear, and its Abstract seems to state sufficiently clearly the main aims of the work. The keyword list seems appropriate. Section 1 cites some references, but it does not provide a sufficiently exhaustive overview and critical discussion of the state of the art of the related literature. As further remark, usually the end of Section 1 should have summarised the general structure of the manuscript by briefly listing the contents of its sections. Section 2 is extremely short, and it could have been included in Section 3. Section 3 should have addressed more details regarding the considered models and tools; in particular, it does not consider the robustness and reliability issues, due for example to uncertainty and disturbance effects, as well as the model-reality mismatch. This point is fundamental when the reliability and robustness features of the proposed solutions have to be verified and validated with respect to real engineering and safety critical systems. Therefore, the effectiveness of the methodology proposed in Section 4 remains unclear and questionable. The authors should have helped the reader to understand the novelty issues of the developed scheme. Due to these flaws, the results considered in Sections 4 and 5 do not help the reader to understand the effectiveness and the efficacy of the proposed solutions. Moreover, more effective metrics and performance indices should be exploited to assess the advantages of the developed techniques. Finally, Section 6 does not suggest effective open problems and future issues that could require further investigations. On the other hand, the use of too many details and achievements should have been avoided here, as it should remain a stand-alone part of the manuscript.
Author Response
In the reviewer’s opinion, the paper could have been more interesting and better organised. In general, the overall contribution remains scientifically poor and technically questionable. In more detail, the paper’s title is quite clear, and its Abstract seems to state sufficiently clearly the main aims of the work. The keyword list seems appropriate. Section 1 cites some references, but it does not provide a sufficiently exhaustive overview and critical discussion of the state of the art of the related literature.
Response:
Additional references have been added to further cover the literature and the following paragraph was added to describe the state of the art:
There is a strong theoretical foundation for quantifying the value of a grid resilience investment, despite challenges that remain to instantiate these methods within utility planning practice. Rickerson et al. overview a host of analytical practices to value the resilience impact of distributed energy resources (DERs), along with an evaluation of established methods using four criteria [9]. The established methods reviewed are the contingent valuation method used by the Interruption Cost Estimator (ICE) tool, the damage cost method within the Federal Emergency Management Agency’s Benefit Cost Analysis Toolkit, and the input-output methodology within tools such as IMPLAN and REMI. The synthesis shows that none of these methods capture the benefits of avoiding especially long-duration outages, and that these approaches do not reflect the full scope of benefits that electric utility regulators may value. Sanstad discusses the theoretical underpinning of several of these methodologies, while providing research directions that may overcome the gaps identified by Rickerson et al. Recent efforts by Baik et al. as well as Ericson and Lisell have proposed a hybrid approach that combines several of the concepts overviewed by Sanstad, and offers a clear path forward to calibrating and validating these models. Case studies that directly compare these various approaches and describe the findings using language and concepts that are readily accessible to electric utility regulators were not found in the literature.
As further remark, usually the end of Section 1 should have summarised the general structure of the manuscript by briefly listing the contents of its sections.
Response:
As suggested, we have added a breakdown of the manuscript as is traditional convention:
The paper is organized into five sections. This introduction has introduced the need for methodologies for quantification of the resiliency value of grid investments such as microgrids and the application of such methods to practical problems. Section 2 presents the proposed valuation methodology for resiliency-enhancing grid investments. Section 3 describes the Vieques Culebra case study and the microgrid DER sizing methodology and costs. Section 4 presents the conceptual design for the microgrid considering the microgrid would be composed of utility-owned and customer owned assets. The authors present conclusions as Section 5.
Section 2 is extremely short, and it could have been included in Section 3.
Response:
Section 2 was combined with what was previously Section 4 Microgrid Sizing and Sensitivity Analysis as what is now subsection 3.1. As Section 3 (now Section 2) introduced the valuation approach, it was considered more applicable to combine the description of the islands with Section 3 and rename Microgrid Case Study.
Section 3 should have addressed more details regarding the considered models and tools; in particular, it does not consider the robustness and reliability issues, due for example to uncertainty and disturbance effects, as well as the model-reality mismatch. This point is fundamental when the reliability and robustness features of the proposed solutions have to be verified and validated with respect to real engineering and safety critical systems. Therefore, the effectiveness of the methodology proposed in Section 4 remains unclear and questionable. The authors should have helped the reader to understand the novelty issues of the developed scheme. Due to these flaws, the results considered in Sections 4 and 5 do not help the reader to understand the effectiveness and the efficacy of the proposed solutions. Moreover, more effective metrics and performance indices should be exploited to assess the advantages of the developed techniques. Finally, Section 6 does not suggest effective open problems and future issues that could require further investigations. On the other hand, the use of too many details and achievements should have been avoided here, as it should remain a stand-alone part of the manuscript.
Response:
As per the Reviewer’s suggestion we have discussed the future problems requiring further study and the details have been softened here. The revised conclusion is presented here:
This research work presented the economic resilience performance of a multi-island electric utility during hurricanes Irma and Maria, calculated for the islands of Vieques and Culebra, through resilience curves and a probabilistic risk-based resilience valuation methodology. The resilience curves suggest it took more than three months to completely restore Vieques. When using IEEE suggested resilience metrics [4], this would result in a storm event X near 100%, which is a total failure. Given how devastating hurricane Maria was, it is difficult to assert the fragility of the system pre-disaster.
To enhance resilience, a microgrid solution is proposed, the cost of which was measured against the societal impact of the two hurricanes. The proposed microgrids will rely on 10MWh BESS and 15 MW PV generation and will provide up to 42% reduced in diesel consumption during loss of the subtransmission supply and up to 56% reduction in annual energy consumption of the minigrid of Vieques and Culebra from the main island during blue sky operation.
The economic analysis suggests the proposed microgrid projects can add between $1.9 and $20 million of expected annual avoided loss by improving reliability and resilience of the discussed locations, by supporting the community in future major weather events. The scalable layered architecture under consideration allows scaling to larger systems, or as in the case of the proposed approach, build out resilient sublayers of diesel microgrid as renewables are integrated through VPP or other energy procurement exercises. As the team plans implementation of the proposed microgrid, it is expected the proposed innovations will benefit underrepresented and vulnerable communities and the industry at large. Furthermore, as the parameters of the resilience valuation methodology are validated, it can serve as a basis for future cost-benefit analysis requirements from policies that encourage electrical system resilience investment.
As shown by the direct comparison within a single case study of four different economic resilience valuation methods, consideration of non-linear effects due to compounding economic conditions over the dynamic evolution of especially long-duration power outages leads to a potentially different valuation of resilience benefits for a resilience investment that reduces outage risk. However, this case study did not validate the parameters for the time-varying customer damage function approach, and instead presents three illustrative parameterizations. While theoretically the time-varying customer damage function method is more precise than the three other methods presented, this study shows that obtaining valid parameters setting the shape of these damage functions is critical, since different parameterizations lead to valuations of resilience investments that span a wide range. Future work is necessary to achieve this validation, as outlined by Baik et al. [11]. Furthermore, while a deterministic approach was illustrated to determining the probability of occurrence at different outage durations, in reality these values are highly uncertain. Future work should perform more rigorous uncertainty quantification and sensitivity analysis to describe the range of likely valuations for grid resilience investments.
Reviewer 3 Report
This paper presents a technical-economic study highlighting the microgrid as a solution for avoiding blackouts due to hurricanes in two islands near Puerto Rico. This study is complete and the results are convincing.
However, I have some reserves concerning the operation of the microgrid in such extreme meteorological conditions. Can we establish fastly the necessary wireless connection for the control?
will all the controllers work correctly? if we don't have sufficient irradiation after the hurricane, the microgrid will not be able to provide power since the PV is the only source with the diesel generator.
I suggest adding an additional cost for the special security of the microgrid elements during the hurricane episode
Author Response
This paper presents a technical-economic study highlighting the microgrid as a solution for avoiding blackouts due to hurricanes in two islands near Puerto Rico. This study is complete and the results are convincing.
However, I have some reserves concerning the operation of the microgrid in such extreme meteorological conditions. Can we establish fastly the necessary wireless connection for the control?
will all the controllers work correctly? if we don't have sufficient irradiation after the hurricane, the microgrid will not be able to provide power since the PV is the only source with the diesel generator.
Response:
These are valid considerations and will need to be part of the initial phases of the project and leverage other work being done on the island. The following text has been added to the conceptual design section to address the reviewer’s comment.
Resilient operation of the microgrid following major events will also depend on the availability of both DER resources and communications for data exchange between different microgrid components. The BESS will provide the appropriate energy during and immediately following the event together with the diesel generators, when PV generation may be non-existent or minimal. A couple of hours following the event, available PV generation will then replace the diesel generation and can provide significant diesel offset should outage of the subtransmission connection persist for multiple days or weeks. With regards to communication network resilience, cellular, meshed network and even low-earth orbit (LEO) satellite communication options are being considered for redundant communication paths. These technical details will need to be fully clarified as part of the design, build and test phases of the project.
I suggest adding an additional cost for the special security of the microgrid elements during the hurricane episode
Response:
Considering the application and the remoteness of the island, significant contingency has been added to the cost estimates to capture, among other elements, security of the microgrid components. However, the following text was added to the end of the conceptual design section to address the reviewer’s comment.
Finally, as with all critical infrastructure, the microgrid and participating DER will need to be appropriately secured to ensure resiliency to physical or cyberattacks during both normal operation and black sky days. These measures may imply additional costs to the project and will be considered as part of the final design.
Round 2
Reviewer 2 Report
The authors have sufficiently improved the quality of the revised manuscript.